# CRISPR/Cas9-Mediated Disruption of *lrp6a* Leads to Abnormal Median Fin Development and Somitogenesis in Goldfish (*Carassius auratus*)

**DOI:** 10.3390/ijms26157067

**Published:** 2025-07-22

**Authors:** Huijuan Li, Rong Zhang, Xiaowen Wang, Lili Liu, Zhigang Yao, Hua Zhu

**Affiliations:** Beijing Key Laboratory of Fishery Biotechnology, Fisheries Science Institute, Beijing Academy of Agriculture and Forestry Sciences, Beijing 100097, China; lihuijuan@baafs.net.cn (H.L.); zhangrong@baafs.net.cn (R.Z.); wangxiaowen@baafs.net.cn (X.W.); liulili@baafs.net.cn (L.L.); yaozhigang@baafs.net (Z.Y.)

**Keywords:** CIRSPR/Cas9, *lrp6a*, *Carassius auratus*

## Abstract

In this study, we demonstrated that *lrp6a*, a co-receptor in the Wnt signaling pathway, is essential for proper median fin formation and somitogenesis in goldfish. We analyzed the gene’s sequence features and expression patterns in both wen-type and egg-type goldfish, uncovering distinct tissue-specific expression differences between the two varieties. To explore the functional role of *lrp6a*, we performed CRISPR/Cas9-mediated gene knockout using eight designed single-guide RNAs (sgRNAs), of which four showed effective targeting. Three high-efficiency sgRNAs were selected and co-injected into embryos to achieve complete gene disruption. Morphological assessments and X-ray microtomography (μCT) imaging of the resulting mutants revealed various abnormalities, including defects in the dorsal, caudal, and anal fins, as well as skeletal deformities near the caudal peduncle. These results confirm that *lrp6a* plays a key role in median fin development and axial patterning, offering new insights into the genetic regulation of fin formation in teleost fish.

## 1. Introduction

The emergence of appendages, such as fins, limbs, or wings, represents a landmark evolutionary innovation in vertebrates. Among vertebrates, bony fish constitute the most diverse group and are characterized by both paired fins (e.g., pelvic and pectoral fins) bilaterally along the ventral–lateral body wall, and unpaired median fins (e.g., dorsal, caudal, and anal fins) positioned along the trunk midline of the anteroposterior body axis [1]. While the position, number, and morphology of paired and median fins vary significantly across bony fish species, their structure consistently consists of a proximal endoskeletal component (pterygiophores) and a distal dermal skeleton (lepidotrichia), which includes fin rays [2,3,4]. Paired fins and limbs exhibit a high degree of homology, which accounts for the predominant focus of research on paired fins [5]. However, median fins predate paired fins by approximately 100 million years in evolutionary history, and the fossil record indicates that paired fins evolved from median fins, positioning median fins as the most primitive form of appendage [6]. While elucidating the genetic mechanisms underlying median fins is critical for advancing our understanding of appendage development in vertebrates, research in this domain remains insufficient.

The median fins—dorsal, caudal, and anal—of adult fish are positioned along the body’s midline, originating from a continuous fin fold during embryonic or larval stages, known as the larval median fin fold (LMFF) [7,8]. The median fin fold (MFF) hypothesis proposes that the discrete distribution of these fins evolved through the reduction of certain segments in this continuous fold of basal chordates. Specifically, during larval development, apoptosis occurs in specific regions of the LMFF, leading to the reduction of the fin fold in these areas, while the remaining parts develop into the future fin structures [9,10]. However, this hypothesis lacks support from cellular and molecular developmental data. Some studies suggest that the fin fold is a transient structure during development, with its reduction not contributing to the establishment of the dorsal fin primordium [11,12]. Instead, this reduction occurs alongside body growth after the formation of the fin primordium, rather than through the active removal of non-fin-forming regions via cell death. Additionally, the emergence and proliferation of specific mesenchymal cells promote the formation of the dorsal fin primordium [12]. These findings challenge the traditional understanding of median fin development and highlight the need for further exploration of the cellular and molecular mechanisms involved.

Evolutionary homology underpinned the shared molecular mechanisms governing paired fins and limbs, which expressed numerous common regulatory factors [13,14]. Evidence suggested that paired fins derived from median fins, and both structures exhibited a comparable developmental mechanism. Moreover, regulatory factors and signaling pathways classically associated with limb development were found to influence median fin formation [15]. For instance, the T-box transcription factor *Eomesa*, critical for limb patterning, was also indispensable for dorsal fin development in zebrafish, where nonsense mutations caused developmental delays and the absence of the dorsal fin [16]. Likewise, the limb-specific enhancer ZRS played a crucial role in regulating Shh expression during normal limb development [16]. In medaka, mutations in ZRS disrupted Shh expression, leading to malformations in both pectoral and dorsal fins [4]. CRISPR/Cas9-mediated knockout of hox genes in zebrafish and medaka generated mutants exhibiting dorsal or anal fin loss, stepwise posterior extension, or vertebral defects, phenotypes linked to *hoxc* cluster members that spanned regulatory domains governing fin positioning along the anterior–posterior axis [17]. The Fgf signaling pathway, essential for limb bud initiation, contributed to zebrafish fin fold formation [1]. Additionally, a reciprocal activation loop between Fgf and Shh signaling was further identified in the developing dorsal fin of *Ictalurus punctatus* [18].

Reverse genetics served as a cornerstone methodology for elucidating gene functions in developmental biology. Although CRISPR/Cas9-mediated knockout of median fin genes in zebrafish (*Danio rerio*) and medaka (*Oryzias latipes*) generated fin-deficient mutants, these frequently caused embryonic lethality or locomotor deficits, limiting functional characterization [19]. Goldfish (*Carassius auratus*) provided an alternative model system, having developed heritable fin variants (e.g., dorsal fin loss, caudal duplication) through millennia of artificial selection [20,21]. Unlike most teleosts where dorsal fin absence impairs survival, goldfish stably maintained this trait, as evidenced by taxonomically distinct “egg-type” (dorsal finless) and “wen-type” (dorsal fin-intact) lineages [22]. Prior genomic investigations into goldfish dorsal fin loss employed hypothesis-driven approaches, identifying candidate genes (*lrp6a*, *dhfr*, *nrxin*) via whole-genome resequencing-derived SNP comparisons between wen- and egg-type lineages [19]. Subsequent functional validation in zebrafish demonstrated that *lrp6a* deficiency perturbs larval fin fold morphogenesis, while ectopic expression of *Dkk1* during post-fin fold stages abrogates adult dorsal fin formation [19]. These findings implicated Wnt-Lrp6 signaling as a critical regulator of median fin development. In human cells, Xenopus, and zebrafish, LRP6 functioned as a key Wnt coreceptor whose cell surface stability and signalosome assembly were regulated by the USP46 complex through deubiquitylation, thereby modulating Wnt signaling activity essential for tissue homeostasis and development [23]. In zebrafish, LRP6 acted as a Wnt co-receptor that, together with Frizzled proteins and Cachd1, regulated Wnt pathway activity to control neurogenesis, neuronal identity, and left–right asymmetry in the habenula [24]. LRP6 and its paralog LRP5 functioned as Wnt co-receptors by binding Wnt ligands together with Frizzled receptors, undergoing phosphorylation to recruit Dishevelled and assemble signalosomes for β-catenin stabilization, thereby driving transcription of downstream target genes; loss-of-function lrp6 mutants exhibited shortened and malformed fin rays, paralleling findings in LRP5 mutants that displayed reduced Wnt activity and abnormal fin regeneration [19]. In contrast, the secreted antagonist Dkk1 bound the extracellular domains of LRP5/6, triggered receptor internalization, and blocked signalosome assembly and downstream Wnt/β-catenin signaling.

The dorsal fin is an essential part of the median fins. In goldfish, naturally occurring dorsal fin-loss mutants provide a unique opportunity to explore the mechanisms governing median fin development. Therefore, we used these mutants as a focal point to investigate the underlying processes shaping median fin formation. We systematically characterized *lrp6a* through multi-dimensional analyses encompassing sequence variation profiling, spatiotemporal expression patterns, and functional perturbation experiments in both wen- and egg-type goldfish. Our findings not only advance the mechanistic understanding of median fin development in this evolutionarily significant species, but also establish a conceptual framework for investigating vertebrate appendage diversification.

## 2. Results

### 2.1. Sequence Analysis of lrp6a Gene

Comparative sequence analysis confirmed that the *lrp6a* gene was conserved between wen-type and egg-type goldfish. Predicted molecular masses were similar, at 180.41 kDa (pI 5.08) for the wen-type and 180.02 kDa (pI 5.10) for the egg-type variant. SignalP 5.0 identified a conserved signal peptide cleavage site at Gly^19^/Leu^20^ in both isoforms, which indicated their role in the secretory pathway (Figure 1I). TMHMM 2.0 analysis detected a single α-helical transmembrane segment spanning residues 1371–1393 (Figure 1II). The homology modeling using SWISS-MODEL (template A0A0G2K0H3.1) produced tertiary structures that exhibited 80.98% sequence identity with the wen-type variant, while the egg-type ortholog maintained complete structural conservation (Figure 1III). Structural annotation revealed twenty LDL receptor YWTD β-propeller domains, four EGF-like calcium-binding domains, and three class A LDL ligand-binding domains (Figure 1IV).

### 2.2. Homology and Phylogenetic Analysis of lrp6a Gene

To assess genetic conservation, we aligned the amino acid sequences of *lrp6a* from wen-type and egg-type goldfish with those of other Cyprinid species (Figure 2). The *lrp6a* sequence alignment between wen-type and egg-type goldfish revealed a high similarity of 99.57%. Furthermore, *lrp6a* exhibited a high degree of conservation among Cyprinid species, with similarity levels ranging from 92.82% to 99.44%, including *Carassius gibelio* (99.44%), *Cyprinus carpio* (95.72%), *Onychostoma macrolepis* (94.19%), *Puntigrus tetrazona* (93.7%), and *Labeo rohita* (92.82%). Phylogenetic analysis indicated that all Cyprinid species clustered together. Notably, the *lrp6a* sequence from egg-type *C. auratus* clustered with *C. gibelio* and was closely related to the wen-type *C. auratus*. *Cyprinus carpio* was also included in this cluster, while *Puntigrus tetrazona* and *Onychostoma macrolepis* formed a separate branch (Figure 3).

### 2.3. Tissue-Specific Expression of lrp6a in Goldfish

The tissue-specific expression of *lrp6a* in adult Oranda and Ranchu goldfish was examined using qPCR across multiple tissues (Figure 4). Distinct expression patterns were observed between the two varieties. In Oranda goldfish, the highest *lrp6a* expression was detected in the gonads (*p* < 0.05), followed by the gills, anal fin, pectoral fin, pelvic fin, caudal fin, and dorsal fin. In Ranchu goldfish, *lrp6a* expression was predominantly detected in the liver, with secondary expression observed in the gills. Relative expression of *lrp6a* was assessed in corresponding tissues of wen- and egg-type goldfish. In the liver, expression was significantly higher in egg-type than in wen-type individuals, whereas in the gonads, wen-type fish showed significantly greater expression than egg-type (*p* < 0.05). No significant differences were detected in other tissues.

### 2.4. Detection of lrp6a-sgRNA Effectiveness

Eight single-guide RNAs (sgRNAs) targeting the *lrp6a* gene (*lrp6a*-sgRNA1 to *lrp6a*-sgRNA8) were designed for CRISPR/Cas9-mediated genome editing. The off-target was used to evaluate their efficiency; each sgRNA was co-microinjected with Cas9 mRNA into single-cell stage embryos (Figure 5). Sanger sequencing confirmed successful mutagenesis for four sgRNAs (*lrp6a*-sgRNA1, -sgRNA2, -sgRNA7, and -sgRNA8), as indicated by overlapping chromatogram peaks near the protospacer adjacent motif (PAM) sites (Figure 6). Quantitative analysis showed mutation efficiencies of 90% (27/30), 86.6% (26/30), 13.3% (4/30), and 100% (30/30) for *lrp6a*-sgRNA1, -sgRNA2, -sgRNA7, and -sgRNA8, respectively (Figure 7I). Regrettably, *lrp6a*-sgRNA3, -sgRNA4, -sgRNA5, and -sgRNA6 were ineffective, showing no multiple peaks near PAM sites of the target sequences. The off-target analysis results for these sgRNAs are presented in Appendix A.

### 2.5. Characterization of Indel Mutations in the Injected Embryos

The mutation types of *lrp6a*-sgRNA1, -sgRNA2, -sgRNA7, and -sgRNA8 were analyzed by ICE analysis (Figure 7II). Four sgRNAs were also able to cause a variety of mutations at the target sites of *lrp6a*. For *lrp6a*-sgRNA1, 98% of the sequenced alleles exhibited indel mutations. Of these, deletions were predominant at 77%, ranging from −1 to −26 base pairs, while insertions comprised 21% (with sizes of 13, 6, and 11 base pairs). For *lrp6a*-sgRNA2, 65% of the alleles showed indel mutations. In this case, deletions accounted for 59%—with sizes of −4, −1, −6, −5, and −14 base pairs—whereas insertions were minimal, constituting only 2% and typically involving a single base pair. For *lrp6a*-sgRNA7, indel mutations were observed in 44% of the sequences. Deletions made up 28% of the total (with sizes of −4, −12, −14, and −5 base pairs), and insertions were noted in 12% of the sequences (with sizes of 11, 1, and 19 base pairs). Finally, for *lrp6a*-sgRNA8, 39% of the sequences exhibited indel mutations. In this group, all mutations were deletions (28%), ranging from −2 to −8 base pairs (Figure 8). Collectively, these results demonstrate that each sgRNA produces a distinct mutation profile, highlighting variability in both the frequency and nature of the indel events.

### 2.6. Identification of Mutate Phenotypes

To enhance the knockout efficiency of *lrp6a* in the F0 generation, we co-injected three sgRNAs (*lrp6a*-sgRNA1, *lrp6a*-sgRNA2, and *lrp6a*-sgRNA8) with demonstrated mutation efficiencies exceeding 80% alongside Cas9 mRNA into embryos, followed by comprehensive phenotypic analysis of mutant individuals. Systematic phenotypic monitoring was conducted at critical developmental stages of −180 and −360 days post-fertilization (dpf) (Figure 9). Quantitative analysis revealed that 31.52% of F0 mutants exhibited varying degrees of structural impairments or complete absence in median fins (including dorsal, caudal, and anal fins) and posterior somite regions. The observed abnormalities manifested four distinct mutation types: 9.2% (115/1250) of mutants displayed specific dorsal fin defects. 7.52% (94/1250) showed combined posterior somite truncation with caudal fin loss, 2.8% (35/1250) demonstrated complete absence of posterior somites, caudal fin, and anal fin, while 12% (150/1250) exhibited isolated caudal fin deficiency. Specially, the severity of dorsal fin defects was quantified using a standardized scoring system based on the proportion of fin area missing. Each individual classified into one of four categories: normal—no dorsal fin loss; mild—area loss less than 50%; moderate—area loss equal to or greater than 50% but not complete loss; or severe—complete absence of the dorsal fin (100% fin loss). Specifically, 4% (50/1250) of individuals were categorized as mild (fin area loss < 50%), 2.4% (30/1250) as moderate (fin area loss ≥ 50% but not complete), and 2.8% (35/1250) as severe (complete dorsal fin loss) (Figure 9).

To systematically evaluate the skeletal consequences of *lrp6a* disruption, high-resolution X-ray microtomography (μCT) was employed to analyze the axial and appendicular skeletal systems in CRISPR-edited specimens (Figure 10). The imaging data corroborated phenotypic observations: individuals exhibiting complete or partial dorsal fin loss displayed concomitant absence of both endoskeletal radials and exoskeletal lepidotrichia. Notably, in specimens with posterior body axis truncation, vertebral column abnormalities were localized to the caudal peduncle region and malformed hemal arches (Figure 10 and Figure 11).

## 3. Discussion

### 3.1. The Sequence Characteristics of lrp6a in Goldfish

Comparative sequence analysis confirmed the high conservation of *lrp6a* between wen-type and egg-type goldfish, with minimal differences in molecular mass and isoelectric points. The structural features identified, including LDL receptor YWTD β-propeller domains, EGF-like calcium-binding domains, and transmembrane regions, suggest that *lrp6a* maintained its canonical role in Wnt signaling across goldfish varieties. The presence of a conserved signal peptide cleavage site further supported its function in the secretory pathway. These findings aligned with previous reports in other vertebrates, where LRP6 acts as a crucial co-receptor in the Wnt/β-catenin pathway, influencing developmental processes such as tissue differentiation and morphogenesis. Phylogenetic analysis revealed that *lrp6a* exhibited high sequence conservation within the Cyprinidae family, clustering closely with *C. gibelio* and *C. carpio*. The minor sequence divergence observed between wen-type and egg-type goldfish likely reflects selective pressures associated with artificial breeding rather than fundamental functional divergence. Notably, the phylogenetic clustering pattern suggested that *lrp6a* function has been evolutionarily conserved across Cyprinid species, supporting its essential role in early development and organogenesis. The 80.98% sequence identity and full structural conservation of *lrp6a* between wen-type and egg-type strains indicated strong functional constraint, consistent with its involvement in fin morphology divergence [19]. Residual amino acid substitutions within conserved domains might have fine-tuned ligand interactions or signaling dynamics, contributing to strain-specific phenotypic variation despite overall conservation.

### 3.2. The Expression Profiles of lrp6a in Goldfish

Tissue-specific expression profiling revealed distinct *lrp6a* expression patterns between wen-goldfish and egg-goldfish varieties. In wen-goldfish specimens, the highest transcript levels were detected in gonadal tissues, whereas egg-goldfish exhibited predominant hepatic expression. This differential expression pattern suggested potential functional divergence in *lrp6a*-mediated regulatory pathways between the two morphological variants. The observed gonadal enrichment in wen-goldfish aligned with established roles of LRP6 orthologs in teleost reproductive system development and germ cell maintenance [25,26,27]. This conservation across evolutionarily distant species reinforced the hypothesis of *lrp6a*’s fundamental involvement in vertebrate reproductive biology. Conversely, the liver-specific expression signature in egg-goldfish implied novel metabolic or developmental functions, potentially associated with hepatic lipid metabolism or organogenesis processes. This finding expanded the recognized functional repertoire of LRP6-related proteins beyond canonical Wnt signaling pathways [28,29]. Additionally, the differential expression might also have resulted from genetic variations in cis-regulatory elements or trans-acting factors accumulated during long-term selective breeding, as well as from potential epigenetic modifications that led to tissue-specific transcriptional regulation.

### 3.3. The sgRNA Efficacy and DNA Repair in Goldfish

The CRISPR/Cas9-mediated mutagenesis analysis revealed critical differences in the efficacy and mutational profiles of *lrp6a*-targeting sgRNAs in goldfish embryos. Among the eight designed sgRNAs, four (*lrp6a*-sgRNA1, -sgRNA2, -sgRNA7, and -sgRNA8) induced detectable mutations, with efficiencies ranging from 13.3% to 100%, while the remaining sgRNAs showed no activity, consistent with the known variability in sgRNA performance due to sequence-specific features such as secondary structures and PAM accessibility [30,31]. The most effective sgRNA (*lrp6a*-sgRNA8) achieved complete mutagenic penetrance, though it predominantly generated small deletions (−2 to −8 bp), whereas *lrp6a*-sgRNA1 produced larger deletions (up to −26 bp) with higher allelic disruption rates (98%). These differences suggested sequence context-dependent variations in DNA repair mechanisms, potentially linked to microhomology-mediated repair biases in teleosts, as previously observed in zebrafish [32]. The observed predominance of deletions over insertions aligned with repair patterns reported in zebrafish but contrasted with mammalian systems, where insertions are more frequent, highlighting species-specific NHEJ dynamics [33]. Overall, the results demonstrated that sgRNA efficiency and mutation profiles varied significantly, emphasizing the importance of optimizing sgRNA selection for effective gene editing in goldfish. Potential off-target effects were assessed during sgRNA design. Off-target sites were predicted using CRISPOR, and sgRNAs with at least four mismatches to any predicted off-target locus were selected to minimize unintended cleavage. Previous studies indicated that sgRNAs do not efficiently bind genomic DNA when mismatches exceed three nucleotides [20,34]. Three sgRNAs (*lrp6a*-sgRNA1, lrp6a-sgRNA2, and *lrp6a*-sgRNA8) met these criteria, each showing ≥4 mismatches at all predicted off-target sites, mostly located in the PAM-proximal region. Off-target cleavage was therefore considered unlikely.

### 3.4. The Role of lrp6a in Median Fin Development

The results demonstrated that *lrp6a* played a critical role in the development of median fins in goldfish, particularly affecting the unpaired fins while leaving the paired fins largely unaffected. This observation supported the hypothesis that, although paired fins originated from median fins and shared common regulatory factors, the median fin represented the most primitive accessory organ with its own dedicated molecular regulatory module [2]. The pronounced sensitivity of median fins to *lrp6a* loss indicated that these fins had retained a conserved Wnt-dependent regulatory module, which likely underpinned the earliest fin-like structures in aquatic vertebrate ancestors. In contrast, paired fins appeared to have gained partial independence from this module by recruiting alternative pathways, thereby facilitating their morphological diversification. This distinction illustrated evolutionary modularity: median and paired fins shared core genetic programs yet diverged in their regulatory control. Furthermore, artificial selection for ornate median fin traits, such as elaborate caudal fins, had likely reinforced *lrp6a*-dependent signaling in these structures, whereas paired fins, being less subject to such selection, had maintained baseline Wnt activity. In addition, the simultaneous absence of dorsal, caudal, and anal fins in certain *lrp6a* knockout individuals suggested that these fins, which initially developed from a continuous fin fold during early embryogenesis, were controlled by overlapping regulatory mechanisms. This finding raised the possibility that additional genes were involved in the coordinated regulation of these median fins, aligning with previous studies that reported similar multi-fin regulatory effects [35].

During whole-genome duplication events, genes generally underwent sub-functionalization, acquiring novel or specialized functions during evolution. In zebrafish, knockout of *lrp6* resulted in fin fold malformations in larvae, subsequently impairing dorsal fin development, with most knockout individuals exhibiting lethality or severe malformations. In the present study, *lrp6a* knockout not only led to dorsal fin defects, but also caused varying degrees of malformation in other median fins, such as the caudal and anal fins. This interspecies functional divergence of *lrp6* might have been related to evolutionary changes in gene function, genome duplication events specific to goldfish, or the strong artificial selection pressure imposed during goldfish domestication. Moreover, unlike zebrafish, *lrp6a* mutant goldfish were viable and could survive to later developmental stages, providing a valuable model for further investigation.

The phenotypic defects observed upon *lrp6a* disruption were found to reflect its central role in modulating Wnt/β-catenin signaling during early embryogenesis. Beyond canonical Wnt activity alone, *lrp6a* was shown to interact with multiple developmental pathways to produce the complex fin phenotypes in fish. Hox gene clusters acted as critical downstream effectors of Wnt signaling for anterior–posterior patterning and fin identity, and loss of *lrp6a* led to altered expression domains of posterior Hox genes, contributing to medially located fin misspecification [36]. In parallel, Sonic hedgehog (Shh) signaling, which normally synergized with Wnt to regulate fin fold morphogenesis and chondrogenesis, was indirectly attenuated in *lrp6a* mutants, exacerbating median fin fold reduction [37]. Furthermore, Fibroblast Growth Factor (FGF) pathways—which formed positive feedback loops with Wnt to maintain the tailbud progenitor pool—were destabilized in the absence of functional *lrp6a*, leading to premature depletion of posterior mesodermal progenitors and simultaneous loss of multiple median fins [38]. Collectively, these findings suggested that *lrp6a* had acted as a pivotal upstream hub within an integrated regulatory network governing median fin development.

### 3.5. The Functions of lrp6a in Axial Skeleton Development and Somitogenesis

The severe truncation of posterior somites and associated axial skeletal elements in *lrp6a* mutants highlighted its essential role in regulating somitogenesis, particularly in the caudal regions. This phenotype mirrored that observed in zebrafish *wnt3a* and *tbx6* mutants, where disruption of Wnt/β-catenin signaling resulted in progressive somite loss toward the tailbud. Mechanistically, *lrp6a* likely modulated the Wnt signaling gradient that governs the segmentation clock, with the posterior-biased severity of the phenotype reflecting reduced Wnt signaling during axial elongation. Additionally, the absence of hemal arches and preural vertebrae in *lrp6a* mutants indicated dual regulatory functions in both osteogenesis and chondrogenesis. These findings align with studies in mammals, where LRP6 was shown to coordinate Wnt signaling with mechanical stimuli during osteoblast differentiation, as evidenced by skeletal malformations in Lrp6-deficient ringelschwanz mice [35,39,40].

The functional conservation of LRP family proteins in skeletal development was further supported by studies on paralogs *Lrp4* and *Lrp5*. In zebrafish, *lrp4* knockdown disrupted caudal and pectoral fin development, along with axial skeletal defects, while murine LRP4 mutations caused limb malformations and syndactyly [41,42]. Similarly, *Lrp5* was shown to regulate bone mineral density and osteoclast activity across vertebrates [43]. The critical involvement of LRP proteins in both fin and limb development underscores their evolutionary conservation in appendage patterning, suggesting that ancestral regulatory modules controlling skeletal morphogenesis were co-opted during the fin-to-limb transition.

## 4. Materials and Methods

### 4.1. Sample Acquisition and Processing

We selected Lionhead goldfish and Ranchu goldfish as representative strains of wen-type goldfish, which have a dorsal fin, and egg-type goldfish, which lack a dorsal fin, respectively. Embryos and larvae at various developmental stages, along with adult tissues, were collected for subsequent analyses. For RNA extraction, embryo–larval samples were preserved in RNA stabilization reagent (RNA-store), incubated at 4 °C overnight, and then stored at −20 °C. For in situ hybridization, specimens were fixed in 4% paraformaldehyde (PFA) at 4 °C overnight, gradually dehydrated through a methanol series, and stored at −20 °C. To inhibit melanogenesis during culture, 3% phenylthiourea (PTU) was added to the aquaculture water for these samples. Embryos older than 24 h post-fertilization (hpf) were enzymatically digested with 2% pronase at room temperature for 20 min, then immediately rinsed in aquaculture water to halt digestion before further cultivation. Adult tissues were snap-frozen in liquid nitrogen and stored at −80 °C for RNA extraction.

### 4.2. Gene Sequence Amplification

Total RNA was extracted from homogenized specimens using TRIzol^®^ reagent (Invitrogen, Waltham, MA, USA) following the manufacturer’s instructions. RNA integrity and purity were assessed via NanoDrop™ 2000 spectrophotometry (Thermo Scientific, Waltham, MA, USA) and 1.5% agarose gel electrophoresis. First-strand cDNA synthesis was performed with PrimeScript™ Reverse Transcriptase (Takara Bio, Kusatsu, Japan) under standardized thermal cycling conditions. To amplify the *lrp6a* (LOC113047914) coding sequence, three primer pairs were designed based on NCBI-annotated genomic data (Appendix A). PCR was conducted using cDNA templates derived from tail fins of wen- and egg-type goldfish. Reactions were performed with 2× Taq II Master Mix under the following cycling conditions: initial denaturation at 95 °C for 3 min, followed by 35 cycles of 95 °C for 30 s, 60 °C for 30 s, and 72 °C for 30 s, with a final extension at 72 °C for 5 min. Amplification products were analyzed via 1.5% agarose gel electrophoresis and verified by bidirectional Sanger sequencing (Sangon Biotech, Shanghai, China).

### 4.3. Bioinformatics Analyses of lrp6a

All sequences were assembled with SeqMan (DNAStar) to generate the complete *lrp6a* cDNA. The open reading frame and corresponding amino acid sequence were predicted using DNAMAN (Lynnon Biosoft). DNAMAN 10 (Lynnon Biosoft) software was used to predict the open reading frame and translate the cDNA sequence to the protein sequence. The isoelectric point (pI) and molecular weight (MW) of the deduced amino acid sequences were predicted using the Compute pI/MW Tool at the ExPAsy site (http://web.expasy.org/compute_pi/, accessed on 2 April 2025). The TMHMM server v.2.0 (https://services.healthtech.dtu.dk/services/TMHMM-2.0/, accessed on 2 April 2025) was used to predict transmembrane helices. The signal peptide and three-dimensional structure of the protein were analyzed with the SignalP 5.0 (https://services.healthtech.dtu.dk/services/SignalP-5.0/, accessed on 2 April 2025) and SMART services (http://smart.embl-heidelberg.de/, accessed on 2 April 2025). The ClustalW 2.1 (Lynnon Biosoft, Los Angeles, CA, USA) was used to align the protein sequences and modified by the ESPript 3.0 (Robert and Gouet, 2014). A phylogenetic tree was constructed by neighbor-joining analysis with 1000 bootstrap replicates using MEGA 11.0 software [44].

### 4.4. Expression Analysis of lrp6a by Real-Time Quantitative PCR

Quantitative real-time PCR (qRT-PCR) was performed to detected the expression patterns of *lrp6a*. Primers were provided in Appendix A, and elongation factor 1-α (EF1-α) was used as the internal control for adult tissues. Complementary DNA (cDNA) was synthesized using the HiScript III 1st Strand cDNA Synthesis Kit (Vazyme). qPCR amplification was carried out with ChamQ SYBR Color qPCR Master Mix (Vazyme) on a Thermo Fisher ABI 7500 system using the following cycling conditions: initial denaturation at 95 °C for 30 s; 40 cycles of 95 °C for 5 s, 60 °C for 20 s, and 72 °C for 20 s. Data were collected from ten biological replicates. Relative expression levels were quantified using the 2^−ΔΔCT^ method. Data analysis and statistical evaluations were performed with GraphPad Prism 10 software to determine significance.

### 4.5. Preparation of sgRNA and Cas9 mRNA

Exons 2 and 3 of *lrp6a* were selected for CRISPR/Cas9-mediated mutagenesis. Target sites were identified using the CRISPOR online tool (http://crispor.tefor.net/, accessed on 2 February 2025), and eight sgRNAs (*lrp6a*-sgRNA1 to *lrp6a*-sgRNA8; Figure 5) were designed accordingly. Potential off-target effects were evaluated during sgRNA design to ensure editing specificity. Off-target sites were predicted using the CRISPOR web tool, and candidate target sequences exhibiting at least four base mismatches to any predicted off-target locus were chosen to minimize unintended editing. DNA templates for sgRNA synthesis were generated via PCR using the DR274 plasmid as a template with target-specific forward primers and a universal reverse primer (Appendix A). PCR conditions were as follows: initial denaturation at 98 °C for 30 s; 35 cycles of 98 °C for 10 s, 60 °C for 20 s, and 72 °C for 20 s; followed by a final extension at 72 °C for 5 min. PCR products were purified using the SanPrep Column PCR Product Purification Kit (Sangon Biotech). sgRNAs were subsequently produced by in vitro transcription using the MEGAshortscript T7 Transcription Kit (Thermo Fisher Scientific, Waltham, MA, USA), purified by phenol-chloroform extraction, and stored at −80 °C.

The Pt3.Cas9-UTRglobin plasmid was linearized with Xba I (NEB, Ipswich, MA, USA) and purified using the MinElute PCR Purification Kit (Qiagen, Hilden, Germany). The linearized plasmid served as a template for in vitro transcription of capped Cas9 mRNA with the T3 RNA Polymerase Kit (Ambion, Waltham, MA, USA). The resulting Cas9 mRNA was purified by phenol-chloroform extraction.

### 4.6. Embryo Microinjection

The injection mixture, containing Cas9 mRNA (300 ng/μL), sgRNA (250 ng/μL), and working buffer (0.5% phenol red, 20 mM HEPES, and 150 mM KCl), was prepared and co-injected into the animal pole of one-cell stage embryos using the NANOLITER2020 system (WPI). Initially, individual sgRNAs were co-injected with Cas9 mRNA to assess targeting efficacy. Subsequently, high-efficiency sgRNA pairs were co-injected with Cas9 mRNA to achieve robust phenotypic penetrance. Approximately 5 nL of the sgRNA/Cas9 solution was injected into each embryo, generally 15–20 min after fertilization, and continued for about 30 min until the first cell division. After injection, embryos were maintained in filtered fresh water at 23 °C. Twelve hours post-injection, poor-quality embryos were manually removed, and the hatched larvae were fed Paramecium and harvest shrimp.

### 4.7. Sanger Sequencing-Based Mutation Screening

For mutation screening, injected embryos were collected at 24 h post-fertilization (hpf). Thirty embryos per sgRNA were randomly selected for genomic DNA extraction using a standard Proteinase K lysis protocol. Briefly, embryos were incubated in 20 μL of DNA extraction buffer at 100 °C for 5 min, cooled on ice for 2 min, then supplemented with 1 μL of Proteinase K (10 mg/mL) and incubated at 55 °C for 2 h, followed by a final incubation at 100 °C for 5 min. The genomic region flanking the target site was amplified using 2× Taq Plus Master Mix II (Dye Plus, Vazyme, Nanjing, China) with primers listed in Appendix A, according to the manufacturer’s instructions. The PCR products were purified and sequenced by Sanger sequencing. Mutation locations and frequencies were determined using ICE analysis (https://ice.editco.bio/#/; accessed on 2 February 2025), which evaluates mutations based on chromatogram data.

### 4.8. Phonotype Screening

Longitudinal observations of median fin phenotypes (caudal, dorsal, and anal) in edited individuals were conducted at 180 and 360 days post-injection (dpf). The images were captured using a camera equipped with a Leica lens. The skeletal system of 360-day-old fish specimens was analyzed using X-ray micro-computed tomography (μCT). Prior to imaging, samples were fixed in 4% paraformaldehyde (PFA) at room temperature for 48 h to preserve tissue integrity. High-resolution scans were performed using a Revvity Quantum GX2 μCT system (PerkinElmer, Waltham, MA, USA) with the following parameters: X-ray voltage = 50 kV, current = 100 μA, field of view (FOV) = 36 mm, isotropic voxel size = 72 μm, and scan duration = 2 min per specimen. Three-dimensional reconstructions were generated using the manufacturer’s proprietary software (Quantum GX Reconstruction v2.1) with a standardized thresholding algorithm optimized for calcified tissue segmentation.

## Figures and Tables

**Figure 1 ijms-26-07067-f001:**
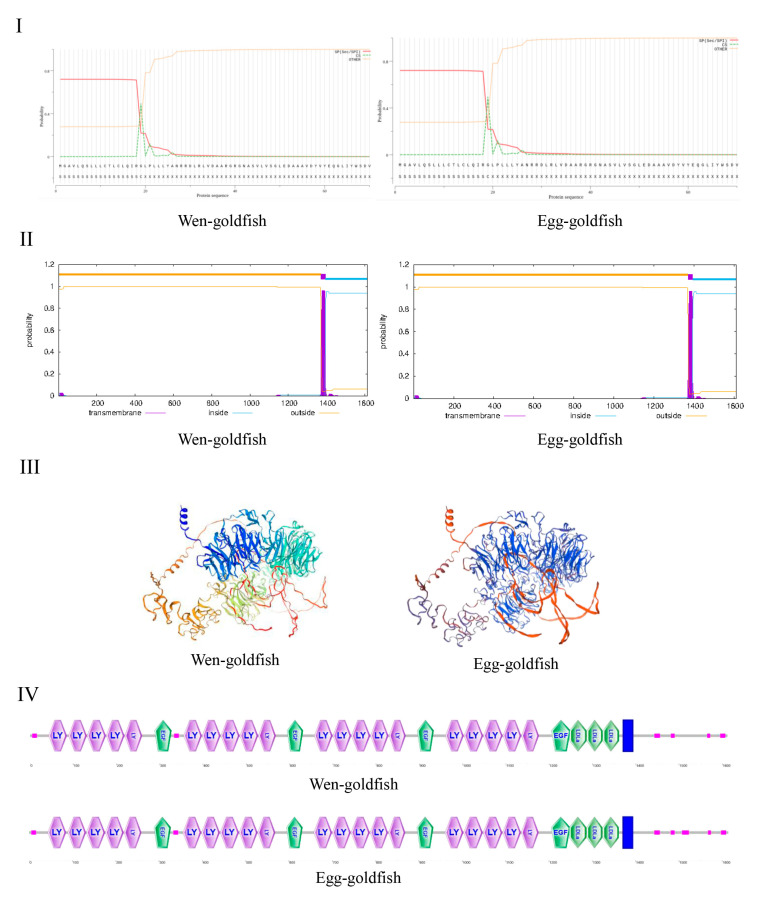
(**I**) represents the signal peptide of the LRP6A protein. (**II**) corresponds to the transmembrane helices of the LRP6A protein. (**III**,**IV**) represent the three-dimensional structure and the conserved motif of the LRP6A protein, respectively.

**Figure 2 ijms-26-07067-f002:**
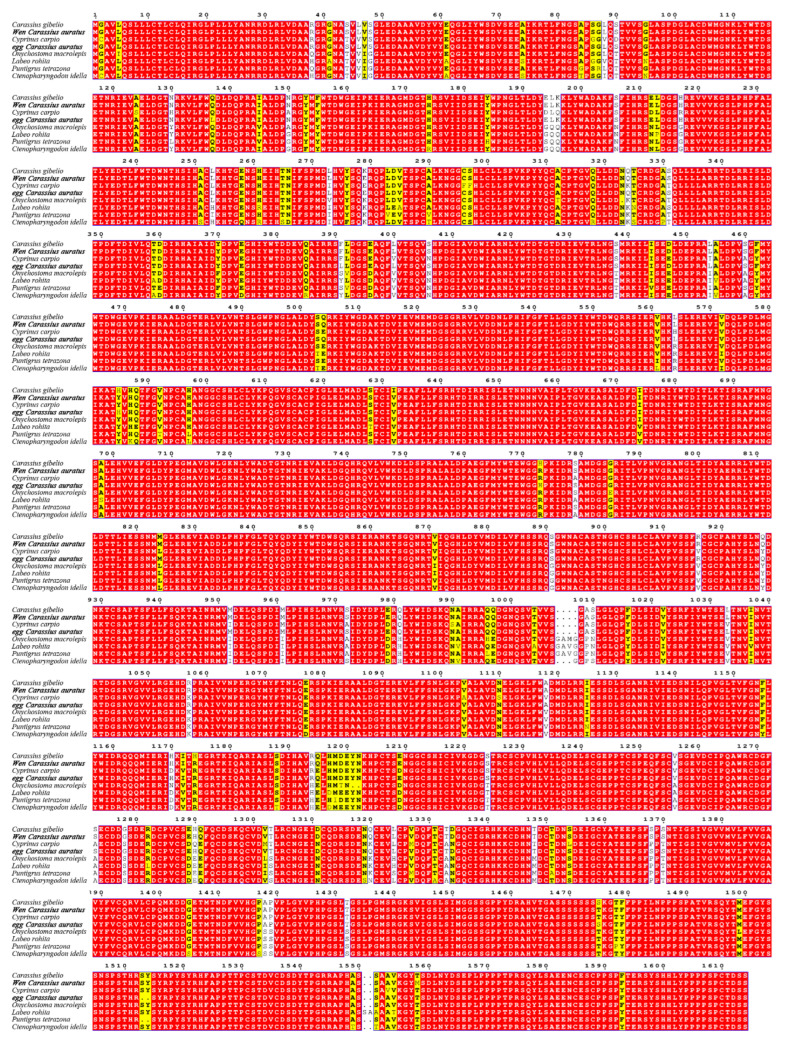
Multiple sequence alignment of the *lrp6a* amino acid sequence among Cyprinid species. The red highlights denote complete identity in amino acid sequences, whereas the yellow highlights indicate partial inconsistency in nucleotide sequences.

**Figure 3 ijms-26-07067-f003:**
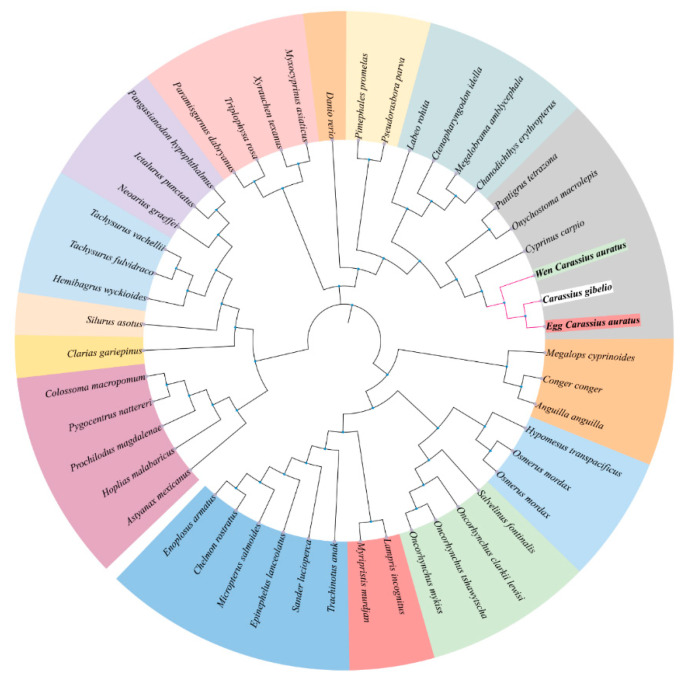
Neighbor-joining phylogenetic tree of *lrp6a* protein.

**Figure 4 ijms-26-07067-f004:**
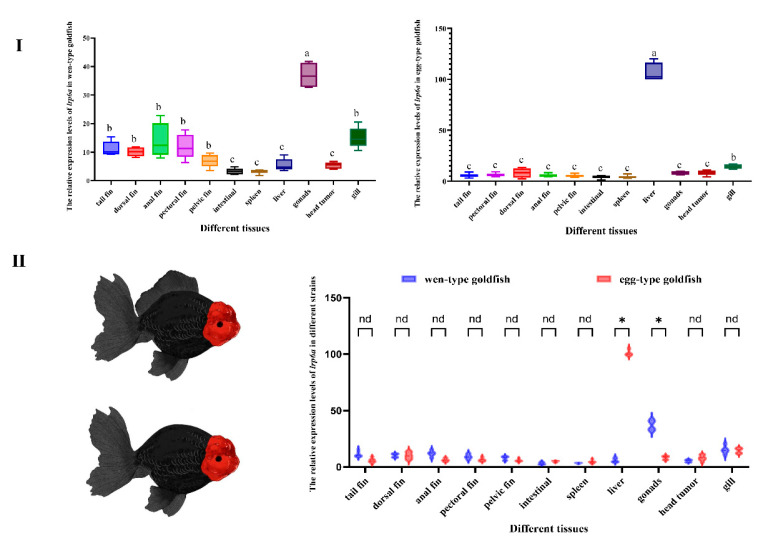
(**I**): Relative expression levels of *lrp6a* across different tissues within each strain. (**II**): Comparative analysis of *lrp6a* expression levels between the two strains in the same tissues. (*p* < 0.05). Letters (a, b, c, etc.) denote significant differences in the relative expression levels of the *lrp6a* gene across different tissues. Distinct letters indicate statistically significant differences (*p* < 0.05). The symbol “*” represents significant differences in gene expression (*p* < 0.05). The abbreviation “nd” stands for “no difference”.

**Figure 5 ijms-26-07067-f005:**
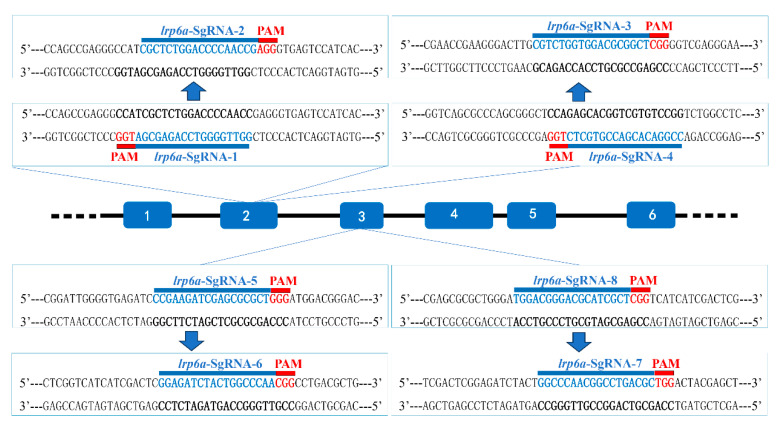
Schematic diagram of *lrp6a*-sgRNAs. The PAMs are shown in red, and the sgRNA sequences are denoted in blue.

**Figure 6 ijms-26-07067-f006:**
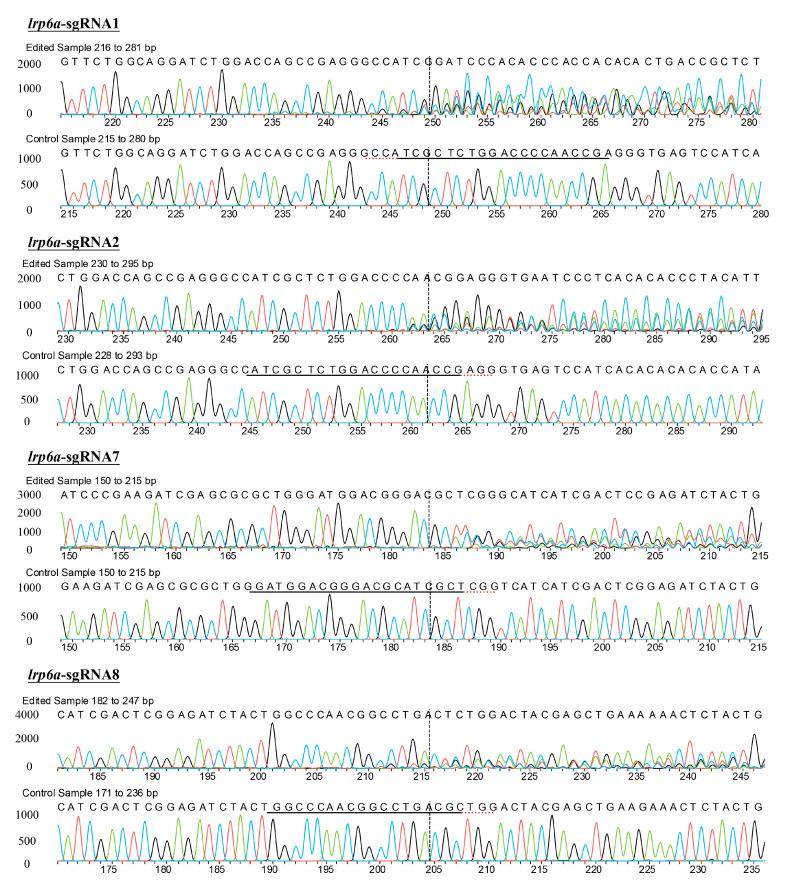
Sanger sequencing of PCR products in the injected embryos. Embryos injected with *lrp6a*-sgRNAs/Cas9 protein were sacrificed for PCR and sequencing. Induced mutations shown by the multiple peaks were detected at the respective target sites of *lrp6a*-sgRNAs. WT means the wild-type.

**Figure 7 ijms-26-07067-f007:**
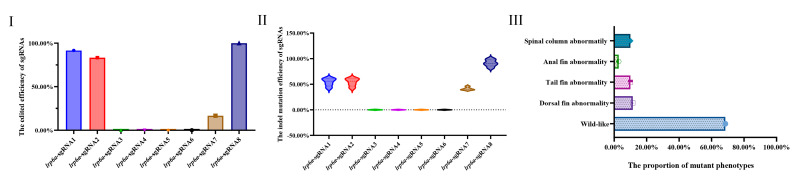
(**I**) shows the editing efficiency of sgRNAs targeting *lrp6a*; (**II**) presents the proportion of indel mutation types for each sgRNA analyzed by ICE; (**III**) illustrates the phenotype rate of edited individuals induced by Cas9/*lrp6a*-sgRNAs.

**Figure 8 ijms-26-07067-f008:**
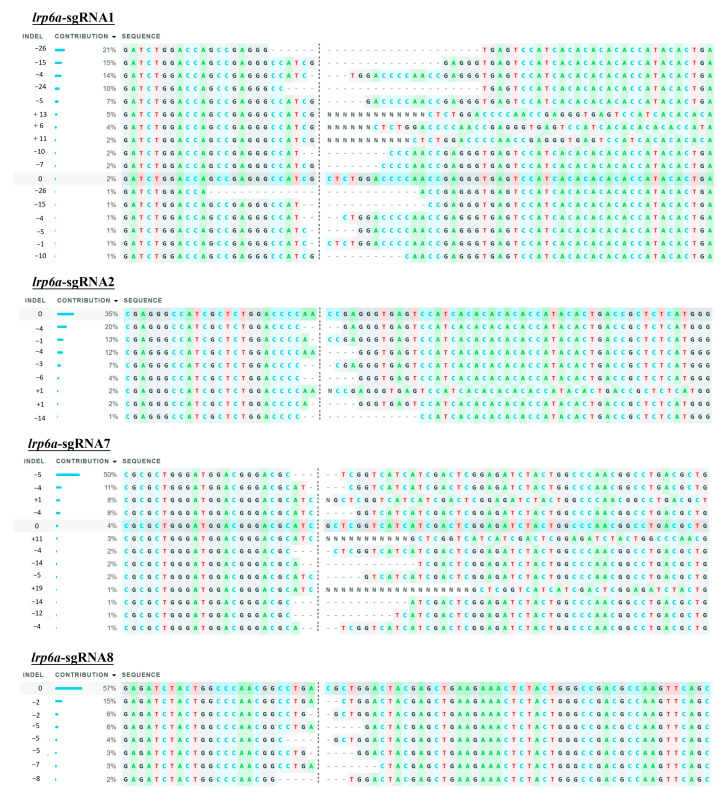
The indel type of *lrp6a*-sgRNAs in individuals via ICE.

**Figure 9 ijms-26-07067-f009:**
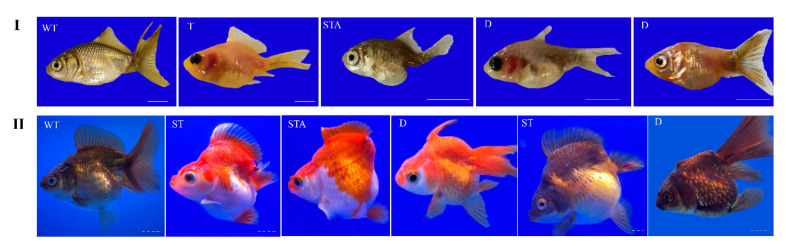
The mutated phenotype in edited individuals. (**I**) represents edited individuals at 180 days post-fertilization (dpf), while (**II**) corresponds to those at 360 dpf. WT (wild-type) indicates unedited control individuals; T (tail) denotes individuals with abnormalities exclusively in the caudal fin; STA (tail-somite-anal fin) represents individuals displaying combined defects in the caudal fin, anal fin, and somites; D (dorsal) refers to individuals with only dorsal fin abnormalities; ST (somite) signifies individuals exhibiting somite-specific defects. The scale bar represented 0.5 centimeters.

**Figure 10 ijms-26-07067-f010:**
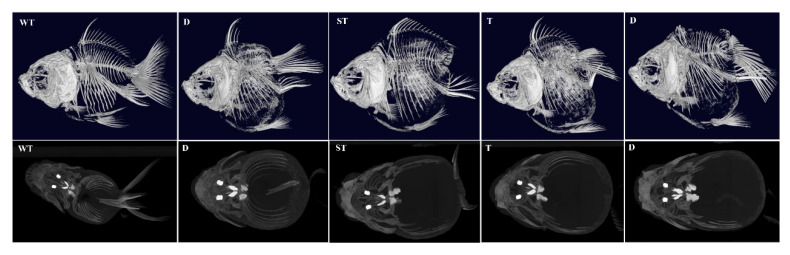
The skeletal system of mutant individuals at 360 days post-fertilization (dpf) was analyzed by the Revvity Quantum GX2 μCT system. WT (wild-type) represents unedited control individuals; T (caudal fin) denotes individuals with abnormalities exclusively in the caudal fin; ST (caudal fin-somite) indicates individuals exhibiting combined defects in the caudal fin and somites; D (dorsal fin) refers to individuals with only dorsal fin abnormalities.

**Figure 11 ijms-26-07067-f011:**
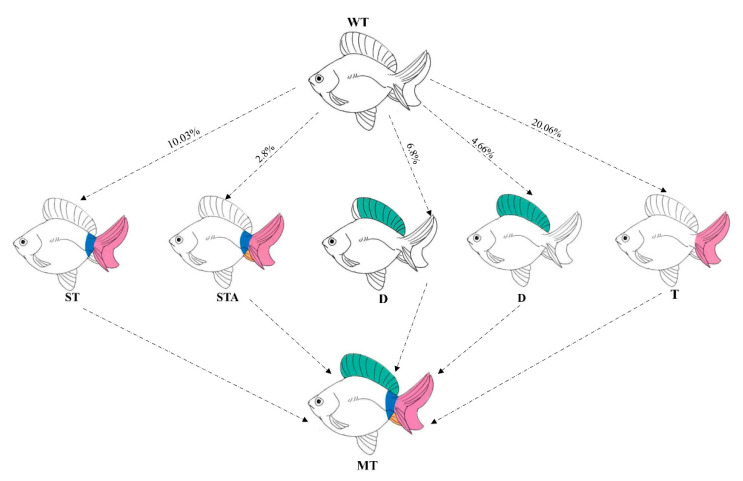
The schematic diagram of phenotypic patterns in edited individuals. WT (wild-type) represents unedited control individuals; T (caudal fin) denotes individuals with abnormalities exclusively in the caudal fin; ST (caudal fin-somite) indicates individuals exhibiting combined defects in the caudal fin and somites; D (dorsal fin) refers to individuals with only dorsal fin abnormalities; MT (mutation) represents the mutation event. The curves in the figure are labeled as the mutation frequencies of the phenotype occurrence.

## Data Availability

The data presented in this study are available on request from the corresponding authors.

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
