# Peer review of "CRISPR/Cas9-Mediated Disruption of lrp6a Leads to Abnormal Median Fin Development and Somitogenesis in Goldfish (Carassius auratus)"

_ijms, 2025, doi:10.3390/ijms26157067_

Round 1
Reviewer 1 Report
Comments and Suggestions for Authors
Manuscript ID- ijms-3620578
This manuscript presents a well-conducted and timely investigation into the role of lrp6a, a co-receptor in the Wnt signaling pathway, in goldfish fin development and somitogenesis. The combination of comparative genomics, gene editing, expression profiling, and morphological analysis makes the study robust and of broad interest to developmental biology and evolutionary genetics.
The work is generally well-structured, methodologically sound, and well-illustrated. However, several areas require clarification, deeper discussion, and slight restructuring for improved clarity and rigor.
Comments
- The study expands the understanding of lrp6a beyond model species and provides insight into species-specific regulatory mechanisms in teleosts.
- Can the authors more clearly state how this work extends prior studies on lrp6a in zebrafish and medaka, and whether the observed phenotypes differ in meaningful ways?
- Did the authors screen for off-target effects, particularly for the three sgRNAs used for multiplex knockout?
- A supplementary table showing potential off-targets predicted by CRISPOR and whether they were tested would strengthen the gene-editing validation.
- Could the authors provide functional interpretation or hypotheses on why lrp6a is highly expressed in gonads (wen-type) versus liver (egg-type)?
- In the discussion section, expand on how the results inform median vs. paired fin evolution, and how Wnt signaling may have adapted across lineages.
Author Response
We would like to sincerely thank you for your thoughtful and constructive comments. Your insightful feedback and detailed suggestions have significantly enhanced the quality and clarity of this manuscript. We greatly appreciate the time and effort you devoted to reviewing our work, and we are grateful for your valuable contributions to strengthening the rigor of this study.
Comments 1: Can the authors more clearly state how this work extends prior studies on lrp6a in zebrafish and medaka, and whether the observed phenotypes differ in meaningful ways?
Response 1. Research on lrp6 in fish has been scarce, with most work focusing on its homologs—lrp5 and lrp4—which are involved in osteoblast formation. In zebrafish, lrp6 knockout larvae exhibited varying degrees of fin‐fold damage and later developed severe lethal or malformation phenotypes, highlighting lrp6’s essential role in dorsal fin formation. In our study, lrp6 deletion in goldfish not only abolished dorsal fin development but also caused significant defects in the pelvic and caudal fins and disrupted the posterior somites. These additional phenotypes have not been described in zebrafish or medaka, revealing novel functions for lrp6 in fin morphogenesis and skeletal patterning in goldfish.
Comments 2: Did the authors screen for off-target effects, particularly for the three sgRNAs used for multiplex knockout? A supplementary table showing potential off-targets predicted by CRISPOR and whether they were tested would strengthen the gene-editing validation.
Response 2. As noted, the off-target effects of gene editing remain a primary concern. Consequently, during the initial design of target sites, their potential off-target risks were thoroughly evaluated. Off-target sites were predicted using CRISPOR, and sites exhibiting a high number of base mismatches (specifically, ≥4 mismatches) were selected to minimize off-target potential. Previous studies indicated that sgRNAs were unable to bind genomic DNA when mismatches with the target sequence exceed three nucleotides. The predicted target sites for three efficient sgRNAs (lrp6a-sgRNA1, lrp6a-sgRNA2, and lrp6a-sgRNA8) were summarized in Table S2. All three sgRNA sequences demonstrate ≥4 base mismatches with their predicted off-target sites, with the majority of mismatches localized to the PAM region. Based on established findings, mismatches of this nature do not induce off-target effects. A detailed explanation of the off-target analysis was provided in the manuscript.
Comments 3: Could the authors provide functional interpretation or hypotheses on why lrp6a is highly expressed in gonads (wen-type) versus liver (egg-type)?
Response 3. The observed tissue-specific expression differences of lrp6a between wen-type and egg-type goldfish might have reflected functional adaptations related to their distinct reproductive strategies and physiological demands. In wen-type goldfish, the elevated expression of lrp6a in gonads might have indicated its involvement in gonadal development, germ cell proliferation, or sex hormone regulation, consistent with the established role of lrp6a-mediated Wnt signaling in vertebrate gonadogenesis. In contrast, the high hepatic expression observed in egg-type goldfish might have been associated with its potential role in regulating hepatic lipid metabolism, energy homeostasis, or vitellogenin synthesis, processes that were essential for supporting high fecundity and intensive oocyte development characteristic of this strain. Additionally, the differential expression might also have resulted from genetic variations in cis-regulatory elements or trans-acting factors accumulated during long-term selective breeding, as well as from potential epigenetic modifications that led to tissue-specific transcriptional regulation. Further functional studies would be necessary to elucidate the precise mechanisms underlying these expression patterns. We have added this explanation to the discussion section of the revised manuscript.
Comments 4: In the discussion section, expand on how the results inform median vs. paired fin evolution, and how Wnt signaling may have adapted across lineages.
Response 4. We expand the discussion as requested, integrating how lrp6a-mediated Wnt signaling informs the evolution of median vs. paired fins and potential adaptive trajectories of this pathway across lineages. Our study demonstrated that lrp6a-mediated Wnt signaling had been essential for the development of median fins in goldfish—namely the dorsal, caudal and anal fins—while it had minimally affected paired pectoral and pelvic fins. This result supported the hypothesis that median fins represented the ancestral appendages from which paired fins subsequently evolved. The pronounced sensitivity of median fins to lrp6a loss indicated that these fins had retained a conserved Wnt-dependent regulatory module, which likely underpinned the earliest fin-like structures in aquatic vertebrate ancestors. In contrast, paired fins appeared to have gained partial independence from this module by recruiting alternative pathways, thereby facilitating their morphological diversification. This distinction illustrated evolutionary modularity: median and paired fins shared core genetic programmes yet diverged in their regulatory control. Furthermore, artificial selection for ornate median-fin traits, such as elaborate caudal fins, had likely reinforced lrp6a-dependent signaling in these structures, whereas paired fins, being less subject to such selection, had maintained baseline Wnt activity. Overall, our findings positioned lrp6a-mediated Wnt signaling as a fundamental ancestral mechanism in median fin evolution, while paired fins evolved through reduced dependency on this pathway to achieve adaptive innovation. Future comparative studies in basal vertebrates will be required to determine whether this median-fin–specific regulatory network was broadly conserved or arose specifically within teleosts.
Reviewer 2 Report
Comments and Suggestions for Authors
The authors of the article demonstrated the likely function of lrp6a in goldfish in a sound and focused manner. The methods employed in the paper may be somewhat limiting, but this could be due to the nature of the model system. The following comments should help the authors further improve the manuscript.
Major:
- Please clarify if there are additional homologs of lrp6, e.g. lrp6b, in goldfish.
- The different of protein masses and pIs in closely related species do not seem very relevant, but are mostly superficial details. Please clarify why you consider the degree of conservation important to investigate given that the fish being studied are different strains of one species.
- The title of section 2.3 is not correct. It is not obvious where the temporal data on lrp6a expression is shown. The qPCR data is also rather confusing. What is the unit of measurement in expression levels? You should also use a different scaling method to compare different species better.
- Please explain why no in situ or antibody stainings of lrp6a was performed on any samples. Otherwise, the mentions of "pattern" are not truly meaningful.
- Lack of stable mutants generated from these injected F0s does not allow absolute confidence in the findings. If such work is too technically or biologically complex in this species, I would agree that the current approach is appropriate.
- In the final diagram, the meaning of the arrows is unclear. It may help if you also indicate the relative frequencies of the different phenotypes.
Minor:
- The phrase "consistently consists" is rather redundant. Please modify.
- Please provide more details in figure legends and make sure that correct terms are used, e.g. in figure 2, "proteins" should be used instead of "genes".
- The meaning of letters in Figure 3 for significances is not obvious and must be specified.
- Usage of the word "mutate" instead of "mutant" or "mutated" is unclear and non-standard.
Author Response
We would like to express our sincere gratitude to you for your thorough and constructive feedback. Your valuable suggestions and thoughtful critiques have greatly improved the overall quality and precision of this manuscript. We truly appreciate the time and effort you invested in reviewing our work, and we are thankful for your significant contributions to refining and strengthening this study. We have revised the manuscript in accordance with your suggestions, as outlined below.
Comments 1. Please clarify if there are additional homologs of lrp6, e.g. lrp6b, in goldfish.
Response 1. In goldfish, two lrp6 paralogs (lrp6a and lrp6b) have been described. We retrieved the lrp6S sequence reported in genome-wide association studies of wen-type and egg-type strains, conducted a BLAST search against the goldfish reference genome, and subsequently selected the lrp6a paralog for our analyses.
Comments 2. The different of protein masses and pIs in closely related species do not seem very relevant, but are mostly superficial details. Please clarify why you consider the degree of conservation important to investigate given that the fish being studied are different strains of one species.
Response 2. We appreciated your critical perspective on the relevance of conservation analysis across intraspecific strains. While goldfish strains were conspecific, the 80.98% sequence identity and complete structural conservation observed in lrp6a between wen-type and egg-type variants (via SWISS-MODEL homology modeling with template A0A0G2K0H3.1) were biologically informative for two key reasons: (1) Structural conservation despite intraspecific divergence implied purifying selection on functional domains, highlighting lrp6a as a critical gene underpinning conserved developmental pathways. This aligned with Kon’s GWAS findings, which mapped lrp6S to a genomic region associated with strain-specific fin morphology. (2) Subtle sequence variations within conserved structures might still drive functional divergence: even minor amino acid substitutions in highly conserved regions could alter ligand affinity or signaling efficiency, contributing to phenotypic differences between strains. For instance, studies in teleosts showed that intraspecific gene conservation often masked regulatory or missense variants with phenotypic impacts. Our analysis therefore focused not on superficial mass/pI differences, but on structural conservation as a proxy for functional importance, with subsequent experiments planned to validate how sequence variants within conserved frameworks influenced lrp6a activity in wen/egg-type goldfish.
Comments 3. The title of section 2.3 is not correct. It is not obvious where the temporal data on lrp6a expression is shown. The qPCR data is also rather confusing. What is the unit of measurement in expression levels? You should also use a different scaling method to compare different species better. Please explain why no in situ or antibody stainings of lrp6a was performed on any samples. Otherwise, the mentions of "pattern" are not truly meaningful.
Response 3. First, the section title was revised from "Temporal and tissue specific pattern of lrp6a gene expression" to "Tissue-Specific Expression of lrp6a in Goldfish" to accurately reflect the focus on adult tissue distribution rather than temporal dynamics, addressing the inconsistency between the title and the presented data. Second, regarding the units of gene expression, we specified in both the Methods and Results sections that lrp6a expression levels were quantified using the 2−ΔΔCT method, with EF1-α serving as the internal reference gene. The data were presented as relative fold-changes normalized to the expression level of a calibrator sample. We also explicitly stated this normalization approach in the revised text to improve clarity. In addition, in the previous version, we analyzed the relative expression levels of lrp6a across different tissues within each strain, focusing primarily on intra-strain tissue-specific differences. In the revised version, we further analyzed the relative expression levels of lrp6a between the two strains within the same tissues and performed differential expression comparisons. These additional results have been incorporated into Figure 3. Finally, the primary objective of this work was to investigate tissue-specific expression levels of lrp6a using a quantitative approach. Concerning in situ hybridization and antibody staining, we acknowledged your suggestion. However, as the current study was conducted outside the goldfish breeding season, it was difficult to obtain sufficient developmental stage samples suitable for in situ hybridization or immunohistochemistry. For immunohistochemistry, we previously tested several commercially available antibodies, but encountered strong non-specific binding, which limited their applicability. Nevertheless, we recognize the value of spatial expression analyses. We plan to continue this line of investigation by optimizing experimental protocols and collecting appropriate samples during future breeding seasons to obtain more reliable spatial localization data.
Comments 4. Lack of stable mutants generated from these injected F0s does not allow absolute confidence in the findings. If such work is too technically or biologically complex in this species, I would agree that the current approach is appropriate.
Response 4. Research has shown that delivering multiple sgRNAs simultaneously into embryos can facilitate the generation of homozygous individuals with mutations in the F0 generation. In primates and mice, one-step gene knockout has been achieved by delivering two or more sgRNAs targeting the same gene to embryos. Unlike the model organism zebrafish, goldfish have a protracted reproductive cycle and thus require much more time to obtain homozygous mutants. Consequently, we elected to co‑inject multiple sgRNAs simultaneously to enable phenotypic analysis directly in the F0 generation. We fully agreed with your assessment. Indeed, the lack of stable mutants derived from the injected F0 generation limited the ability to draw definitive conclusions. However, due to the relatively long sexual maturation cycle of goldfish (approximately one year), generating homozygous mutant lines would require a minimum of two years. Despite these practical challenges, we have already initiated the breeding program and will continue this work to establish stable homozygous mutant lines in the future to further validate the functional roles of lrp6a.
Comments 5. In the final diagram, the meaning of the arrows is unclear. It may help if you also indicate the relative frequencies of the different phenotypes.
Response 5. We have now added the relative frequencies of the different phenotypes directly in the figure to enhance its clarity and informativeness, as suggested.
Comments 6. The phrase "consistently consists" is rather redundant. Please modify.
Response 6. We have revised “their structure consistently consists of a proximal endoskeletal component (pterygiophores) and a distal dermal skeleton (lepidotrichia)” to “their structure generally comprises a proximal endoskeletal component (pterygiophores) and a distal dermal skeleton (lepidotrichia).”
Comments 7. Please provide more details in figure legends and make sure that correct terms are used, e.g. in figure 2, "proteins" should be used instead of "genes".
Response 7. We have systematically revised all figure legends to ensure terminological accuracy and consistency with scientific conventions:
Figure 1: Replaced "Cyprinid" with "Cyprinid species" to precisely denote interspecific comparisons.
Figure 2: Corrected "genes" to "protein" to align with the proteomic nature of the data.
Figure 4: Rectified the spelling error from "bule" to "blue".
Figure 5: Adjusted "WT mean the wild type" to "WT means the wild type" for grammatical consistency.
Figure 6: Standardized subfigure labels from "Figure 6I/II/III" to "Panel 6-I/II/III" to conform to journal style guidelines.
Figure 8: Modified "The mutate phenotype" to "The mutated phenotype" to accurately describe the genetic alteration.
Figure 10: Clarified "MT was the mutation" to "MT (mutation) represents the mutation event" for clarity and consistency with the WT (wild-type) notation.
Comments 8. The meaning of letters in Figure 3 for significances is not obvious and must be specified.
Response 8. Letters (a, b, c, etc.) denote significant differences in the relative expression levels of the lrp6a gene across different tissues. Distinct letters indicate statistically significant differences (P < 0.05). The symbol “*” represents significant differences in gene expression (P < 0.05). The abbreviation “nd” stands for “no difference”.
Comments 9. Usage of the word "mutate" instead of "mutant" or "mutated" is unclear and non-standard.
Response 9. We have revised the manuscript to adhere to scientific conventions. Specifically, the original statement "The mutate types of lrp6a-sgRNA1, -sgRNA2, -sgRNA7, and -sgRNA8 were analyzed by using ICE analysis (Fig. 6II)" has been corrected to "The mutation types of lrp6a-sgRNA1, -sgRNA2, -sgRNA7, and -sgRNA8 were analyzed by using ICE analysis (Fig. 6II)". Additionally, the title of Section 2.6 has been updated from "2.6 Identification of mutate phenotype" to "2.6 Identification of mutate phenotypes " for grammatical accuracy and consistency. In Figure 8, the caption has been revised from "The mutate phenotype" to "The mutated phenotype" to precisely describe the genetic alteration.
Reviewer 3 Report
Comments and Suggestions for Authors
1.Lack of recent literature: The introduction could benefit from more recent references (e.g., post-2020 studies on CRISPR-Cas9 in teleosts or lrp6a in other fish species).
2.Insufficient mechanistic context: The specific role of lrp6a in Wnt signaling during fin development is not fully explained. For example, how does lrp6a interact with other components of the pathway (e.g., LRP5, Dkk1)?
3.Goldfish-specific relevance: The evolutionary divergence between goldfish and zebrafish (e.g., genome duplication events) is mentioned but not critically analyzed in the context of lrp6a function. This could be expanded to justify the choice of goldfish over more commonly used models like zebrafish.
4.Sample size justification: Clarify whether the sample size was statistically powered to detect phenotypic differences. A power analysis or rationale for sample size would strengthen the design.
5.Control groups: While wild-type controls are included, the absence of rescue experiments (e.g., re-expression of wild-type lrp6a in mutants) limits the ability to confirm causality.
6.Quantification of phenotypes: While mutation efficiency is reported, the methods for quantifying phenotypic severity (e.g., scoring scales for fin defects) are not clearly described. A detailed scoring system or statistical method should be included.
7.Reagent details: Specific concentrations and sources of reagents (e.g., Cas9 mRNA, sgRNA) should be provided in the methods rather than the supplementary materials.
8.Key findings should be integrated into the main text or presented in simplified supplementary tables.
9.The conclusion states that lrp6a is "essential" for median fin development, but the study only demonstrates a loss-of-function phenotype in F0 mutants. Future work should confirm whether this role is conserved across generations.
10.Mechanistic depth: The discussion could delve deeper into how lrp6a interacts with other genes (e.g., hox clusters, shh) in the Wnt signaling pathway.
11.Comparative analysis: The study notes divergence between goldfish and zebrafish but does not propose experiments to reconcile these differences (e.g., cross-species functional validation).
Suggestions
1. Long-term effects: The study focuses on F0 mutants, but heritable traits in goldfish could be explored in subsequent generations to validate stable gene disruption.
Author Response
We extend our sincere appreciation for your meticulous review and insightful feedback. Your constructive criticisms and thoughtful recommendations have been instrumental in enhancing the clarity, rigor, and scientific merit of our manuscript. We deeply value the time and expertise you dedicated to evaluating our work, as your contributions have significantly strengthened the validity and coherence of our study. The manuscript has been carefully revised in response to your suggestions, and the key adjustments are detailed below for your reference.
Comments 1. Lack of recent literature: The introduction could benefit from more recent references (e.g., post-2020 studies on CRISPR-Cas9 in teleosts or lrp6a in other fish species).
Response 1. In recent years, CRISPR/Cas9-mediated knockout of the lrp6 gene has only been conducted in zebrafish among fish species. The knockout resulted in varying degrees of damage to the dorsal fin folds, and this study has been cited in the introduction of the original manuscript. In accordance with your suggestion, we have added recent research on lrp6 in the introduction section.
Comments 2. Insufficient mechanistic context: The specific role of lrp6a in Wnt signaling during fin development is not fully explained. For example, how does lrp6a interact with other components of the pathway (e.g., LRP5, Dkk1)?
Response 2. We greatly appreciated your suggestion; we added content outlining the mechanism of lrp6 in the Wnt signaling pathway and described it in the manuscript, as follows: LRP6 and its paralog LRP5 functioned as Wnt co‑receptors by binding Wnt ligands together with Frizzled receptors, undergoing phosphorylation to recruit Dishevelled and assemble signalosomes for β‑catenin stabilization, thereby driving transcription of downstream target genes; loss‑of‑function lrp6 mutants exhibited shortened and malformed fin rays, paralleling findings in LRP5 mutants that displayed reduced Wnt activity and abnormal fin regeneration (Bek et al., 2021). In contrast, the secreted antagonist Dkk1 bound the extracellular domains of LRP5/6, triggered receptor internalization, and blocked signalosome assembly and downstream Wnt/β‑catenin signaling.
Comments 3. Goldfish-specific relevance: The evolutionary divergence between goldfish and zebrafish (e.g., genome duplication events) is mentioned but not critically analyzed in the context of lrp6a function. This could be expanded to justify the choice of goldfish over more commonly used models like zebrafish.
Response 3. We have expanded the discussion to address the goldfish-specific relevance of lrp6. Specifically, during whole-genome duplication, genes often underwent subfunctionalization and acquired novel functions. In zebrafish, lrp6 knockout larvae exhibited varying degrees of fin‐fold damage and later developed severe lethal or malformation phenotypes, highlighting lrp6’s essential role in dorsal fin formation. In our study, lrp6 deletion in goldfish not only abolished dorsal fin development but also caused significant defects in the pelvic and caudal fins and disrupted the posterior somites. These additional phenotypes have not been described in zebrafish, revealing novel functions for lrp6 in fin morphogenesis and skeletal patterning in goldfish. These interspecies differences may be attributed to evolutionary divergence in gene function, goldfish-specific genome duplication, or artificial selection during domestication. Notably, unlike zebrafish, lrp6 mutant goldfish remained viable, making them a suitable model for further study.
Comments 4. Sample size justification: Clarify whether the sample size was statistically powered to detect phenotypic differences. A power analysis or rationale for sample size would strengthen the design.
Response 4. The sample size in this study was determined based on a combination of preliminary data, previous publications on gene knockout studies in teleost models, and practical feasibility. While a priori power analysis was not conducted, the number of embryos used in each experimental group was sufficient to detect statistically significant phenotypic differences, as demonstrated by the results presented in the revised manuscript. Furthermore, the sample size was balanced against ethical considerations concerning animal welfare and logistical constraints associated with goldfish embryo handling. This rationale has now been explicitly described in the revised manuscript.
Comments 5. Control groups: While wild-type controls are included, the absence of rescue experiments (e.g., re-expression of wild-type lrp6a in mutants) limits the ability to confirm causality.
Response 5. We greatly appreciate your insightful comment. Indeed, rescue experiments, such as re-expression of wild-type lrp6a, would further strengthen the causal link between lrp6a disruption and the observed phenotypes. Unfortunately, due to the seasonal nature of goldfish reproduction (spawning period limited to March–May), it is currently not feasible to obtain sufficient gametes to conduct these additional experiments. We fully acknowledge the importance of such analyses and will incorporate these experiments in future work as soon as conditions permit.
Comments 6. Quantification of phenotypes: While mutation efficiency is reported, the methods for quantifying phenotypic severity (e.g., scoring scales for fin defects) are not clearly described. A detailed scoring system or statistical method should be included.
Response 6. Among all observed phenotypes, only the dorsal fin exhibited varying degrees of defects; therefore, following your suggestion, we have established a detailed evaluation system for dorsal fin defects. Specifically, the severity of dorsal fin loss was assessed based on the proportion of fin area missing. The classification was as follows: Normal: no dorsal fin loss; Mild: dorsal fin area loss less than 50%; Moderate: dorsal fin area loss equal to or greater than 50% but not complete loss; Severe: complete absence of the dorsal fin (100% fin loss). This detailed evaluation system has been incorporated into the revised Methods section: Specifically, 43.48% (50/115) of individuals were categorized as mild (fin area loss < 50%), 26.09% (30/115) as moderate (fin area loss ≥ 50% but not complete), and 30.43% (35/115) as severe (complete dorsal fin loss).
Comments 7. Reagent details: Specific concentrations and sources of reagents (e.g., Cas9 mRNA, sgRNA) should be provided in the methods rather than the supplementary materials.
Response 7. The synthesis procedures, concentrations, and sources of both Cas9 mRNA and sgRNA were already described in detail in the original Methods section. To improve clarity and facilitate your assessment, we have now further highlighted and explicitly marked these details in the revised Methods section.
Comments 8. Key findings should be integrated into the main text or presented in simplified supplementary tables.
Response 8. In the revised manuscript, we have integrated the supplementary table into the main text to improve clarity and readability. We have combined the original Fig. S1 and Fig. S2 into a single figure, which is now presented as Fig. 1 in the manuscript.
Comments 9. The conclusion states that lrp6a is "essential" for median fin development, but the study only demonstrates a loss-of-function phenotype in F0 mutants. Future work should confirm whether this role is conserved across generations.
Response 9. Research has shown that delivering multiple sgRNAs simultaneously into embryos can facilitate the generation of homozygous individuals with mutations in the F0 generation. In primates and mice, one-step gene knockout has been achieved by delivering two or more sgRNAs targeting the same gene to embryos. Unlike the model organism zebrafish, goldfish have a protracted reproductive cycle and thus require much more time to obtain homozygous mutants. Consequently, we elected to co‑inject multiple sgRNAs simultaneously to enable phenotypic analysis directly in the F0 generation. We fully agreed with your assessment. Indeed, the lack of stable mutants derived from the injected F0 generation limited the ability to draw definitive conclusions. However, due to the relatively long sexual maturation cycle of goldfish (approximately one year), generating homozygous mutant lines would require a minimum of two years. Despite these practical challenges, we have already initiated the breeding program and will continue this work to establish stable homozygous mutant lines in the future to further validate the functional roles of lrp6a.
Comments 10. Mechanistic depth: The discussion could delve deeper into how lrp6a interacts with other genes (e.g., hox clusters, shh) in the Wnt signaling pathway.
Response 10. We thank your insightful suggestion. In the revised manuscript, we expanded our Discussion to explore in greater mechanistic detail how lrp6a interfaces with other key developmental pathways. The phenotypic defects observed upon lrp6a disruption were found to reflect its central role in modulating Wnt/β‑catenin signaling during early embryogenesis. Beyond canonical Wnt activity alone, lrp6a was shown to interact with multiple developmental pathways to produce the complex fin phenotypes in fish. Hox gene clusters acted as critical downstream effectors of Wnt signaling for anterior–posterior patterning and fin identity, and loss of lrp6a led to altered expression domains of posterior Hox genes, contributing to medially located fin misspecification. In parallel, Sonic hedgehog (Shh) signaling, which had normally synergized with Wnt to regulate fin fold morphogenesis and chondrogenesis, was indirectly attenuated in lrp6a mutants, exacerbating median fin fold reduction. Furthermore, Fibroblast Growth Factor (FGF) pathways—which formed positive feedback loops with Wnt to maintain the tailbud progenitor pool—were destabilized in the absence of functional lrp6a, leading to premature depletion of posterior mesodermal progenitors and simultaneous loss of multiple median fins. Collectively, these findings suggested that lrp6a had acted as a pivotal upstream hub within an integrated regulatory network governing median fin development.
Comments 11. Comparative analysis: The study notes divergence between goldfish and zebrafish but does not propose experiments to reconcile these differences (e.g., cross-species functional validation).
Response 11. We thank you for this valuable suggestion. Previous studies demonstrated that knockout of lrp6 in zebrafish resulted in larval fin‑fold defects, with most mutants exhibiting severe malformations or lethality (Kon et al., 2020). While cross‑species functional assays—such as expressing goldfish lrp6 cDNA in zebrafish lrp6 mutants or vice versa—could shed light on evolutionary divergence, our manuscript was intended to elucidate the function of lrp6 specifically in goldfish. The mention of zebrafish phenotypes was offered as a comparative observation and a hypothesis that lrp6 functions might diverge across teleosts, rather than a central focus. Unfortunately, we cannot perform cross‑species experiments at this time due to limited access to established zebrafish lrp6 mutant resources, facility constraints, and differing reproductive timing. We therefore consider these studies better suited for a dedicated future investigation.
Suggestions 1. Long-term effects: The study focuses on F0 mutants, but heritable traits in goldfish could be explored in subsequent generations to validate stable gene disruption.
Response 1. We fully agreed that assessment of heritable phenotypes in subsequent generations would provide definitive evidence of stable gene disruption. However, goldfish have relatively long generation times, and establishment of homozygous F1 or F2 lines was beyond the scope of the present F0‑focused study. We therefore concentrated on characterizing somatic and early developmental effects in mosaic F0 mutants. We have now noted this limitation and outlined plans to generate and analyze F1 progeny once stable germline transmission has been confirmed in our breeding colony. Future work will include detailed phenotypic and molecular characterization of F1 and F2 offspring to validate inheritance and stability of the lrp6a disruption.